# Selective Arterial Embolization of Renal Angiomyolipomas with a N-Butyl Cyanoacrylate-Lipiodol Mixture: Efficacy, Safety, Short- and Mid-Term Outcomes

**DOI:** 10.3390/jcm10184062

**Published:** 2021-09-08

**Authors:** François-Victor Prigent, Kévin Guillen, Pierre-Olivier Comby, Julie Pellegrinelli, Nicolas Falvo, Marco Midulla, Nabil Majbri, Olivier Chevallier, Romaric Loffroy

**Affiliations:** 1Image-Guided Therapy Center, Department of Vascular and Interventional Radiology, François-Mitterrand University Hospital, 14 Rue Paul Gaffarel, BP 77908, 21079 Dijon, France; francoisv21@gmail.com (F.-V.P.); kevin.guillen@chu-dijon.fr (K.G.); julie.pellegrinelli@chu-dijon.fr (J.P.); nicolas.falvo@chu-dijon.fr (N.F.); marco.midulla@chu-dijon.fr (M.M.); olivier.chevallier@chu-dijon.fr (O.C.); 2Imaging and Artificial Vision (ImViA) Laboratory-EA 7535, University of Bourgogne/Franche-Comté, 9 Avenue Alain Savary, BP 47870, 21078 Dijon, France; pierre-olivier.comby@chu-dijon.fr; 3Department of Neuroradiology and Emergency Radiology, François-Mitterrand University Hospital, 14 Rue Paul Gaffarel, BP 77908, 21079 Dijon, France; 4Department of Nephrology and Renal Transplantation, François-Mitterrand University Hospital, 14 Rue Paul Gaffarel, BP 77908, 21079 Dijon, France; nabil.majbri@chu-dijon.fr

**Keywords:** angiomyolipoma, bleeding, embolization, interventional radiology, cyanoacrylate, glue

## Abstract

Selective arterial embolization (SAE) for renal angiomyolipoma (rAML) is effective to treat or prevent bleeding. We report our experience using a cyanoacrylate–Lipiodol mixture. We performed a single-center retrospective review of all rAMLs embolized with cyanoacrylate glue between July 2014 and June 2020. Demographics, tuberous sclerosis complex (TSC) status, clinical presentation, angiography features, and follow-up data were recorded. Pre- and post-procedure rAML sizes and volumes were estimated from computed tomography (CT) or magnetic resonance imaging (MRI) studies. Kidney function was assessed before and after the procedure. We identified 24 patients (22 females and 2 males, mean age 51 years) treated for 27 AMLs, either prophylactically (*n* = 20) or as an emergency (*n* = 4). Technical success was achieved for 25/27 AMLs; two patients, each with a single AML, required nephrectomy and repeated embolization, respectively. Major complications occurred in three patients and minor complications such as postembolization syndrome in 15 patients. AML volume reduction after embolization was 55.1% after a mean follow-up of 15 months (range, 1–72 months). Factors associated with greater volume reduction were a smaller percentage of fat (*p* = 0.001), larger initial rAML volume (*p* = 0.014), and longer follow-up (*p* = 0.0001). The mean creatinine level did not change after SAE. Embolization of rAMLs with a mixture of cyanoacrylate and Lipiodol is feasible, safe, and effective in significantly decreasing tumor volume.

## 1. Introduction

Angiomyolipoma (AML) is a benign hamartomatous tumor that accounts for 0.3 to 3% of all renal masses [1]. It is composed of varying proportions of fat, dysmorphic vessels, and smooth muscle tissue [2]. In AMLs with macroscopic fat, a density less than −20 HU by computed tomography (CT) or a signal drop on fat-saturated magnetic resonance imaging (MRI) sequences provides the diagnosis, obviating the need for invasive diagnostic investigations [3]. AMLs tend to expand gradually over time [4]. Among AMLs, 80% are sporadic and 20% occur in patients with tuberous sclerosis complex (TSC), an autosomal dominant phacomatosis whose manifestations include the formation of benign tumors. TSC-related AMLs tend to develop in younger patients, are more frequent and more often bilateral, and exhibit faster growth rates (up to 20%/year versus 5% for solitary sporadic AMLs) [3,5,6]. Both sporadic and TSC-related AMLs occur predominantly in females.

Most renal AMLs (rAMLs) are asymptomatic lesions that are discovered incidentally and require no treatment. Pain and hematuria may develop, however, as the tumor grows. More importantly, rupture of the tumor may cause massive retroperitoneal bleeding with life-threatening hypovolemic shock. Reported risk factors for severe bleeding include tumor diameter greater than 4 cm, aneurysms larger than 5 mm, and TSC [7,8]. Prophylactic treatment may be indicated in patients with these risk factors. Treatment modalities include total nephrectomy, nephron-sparing excision by partial nephrectomy or percutaneous ablation (i.e., radiofrequency ablation, cryoablation, or microwave ablation), and percutaneous selective arterial embolization (SAE). Emergency SAE has been shown to effectively control active tumor bleeding [9]. SAE can also be used prophylactically to treat dysplastic vessels and reduce tumor size. This treatment is particularly recommended in patients with TSC, in whom the multiple bilateral rAMLs require a nephron-sparing technique. In addition to high efficacy, advantages of SAE include its minimally invasive nature and the low complication rate [7].

For SAE used to treat rAMLs, various materials can be used, alone or in combination, including polyvinyl alcohol particles, microspheres, coils, microcoils, and liquid agents [10,11,12]. Each of these materials has its own advantages and drawbacks. Among liquid agents, Onyx^®^ (Medtronic, Dublin, Ireland), in which the non-adhesive glue is ethylene vinyl-alcohol copolymer (EVOH) and the radio-opaque agent tantalum powder, has been used in several studies for SAE of rAMLs [13,14]. On the other hand, N-butyl 2-cyanoacrylate (NBCA) is an adhesive liquid embolic agent whose properties differ from those of Onyx^®^. NBCA, first used for embolization in 1975, is marketed as Histoacryl^®^ (B/Braun, Tuttlingen, Germany) and Glubran^®^2 (GEM SRL, Viareggio, Italy) [15]. The glue is diluted in variable proportions with Lipiodol to ensure radio-opacity. The mixture polymerizes rapidly upon contact with ionic media—blood in the case of arterial embolization—producing nearly instantaneous and permanent occlusion by adhesion to the vascular wall, with local inflammation and thrombus formation [16]. The low viscosity of NBCA allows the embolization of distal arteries. Diffusion of the embolic agent can be monitored readily due to the presence of radio-opaque Lipiodol.

Here, our aim was to evaluate the efficacy of SAE, as assessed based on tumor volume reduction, using Glubran^®^2. We also assessed the technical success rate, potential changes in kidney function after SAE, and other safety outcomes. Finally, we sought to identify factors associated with tumor volume reduction.

## 2. Materials and Methods

### 2.1. Study Design and Patients

We conducted a retrospective observational review of consecutive patients with rAMLs who underwent glue embolization at the Dijon-Bourgogne University Hospital between July 2014 and June 2020. We identified the patients by searching our imaging database using the indexing terms “angiomyolipoma”, “embolization”, and “cyanoacrylate”.

Our ethics committee approved the study and waived the requirement for informed patient consent in compliance with French legislation on retrospective studies of anonymized data.

### 2.2. Diagnosis and Tumor Size

Computed tomography (CT) or magnetic resonance imaging (MRI) of the rAMLs was obtained before SAE in all patients. Postprocessing was performed on a syngo^®^.via workstation (Siemens Healthcare, Erlangen, Germany). The presence of macroscopic fat within the tumor and the absence of calcification or necrosis confirmed the diagnosis. Macroscopic fat appeared as hypoattenuating foci (<−20 Hounsfield Units, HU) on unenhanced CT images or as cancellation of a high-intensity signal on T1-weighted MR images with fat saturation [17]. When no macroscopic fat was visualized, a biopsy was obtained.

Tumor size was assessed by measuring the maximum diameter and estimating the three-dimensional 3D volume after manual contouring, both on the last CT or MRI before embolization and on the most recent CT or MRI obtained during follow-up. When CT was performed, the percentage of macroscopic fat was estimating by creating a density histogram using syngo.MM Oncology software (Siemens Healthcare). Aneurysms were measured on diagnostic imaging studies performed with injected contrast material. In patients with more than one aneurysm, the largest one was selected.

### 2.3. Selective Arterial Embolization (SAE)

The decision to perform embolization was made during a multidisciplinary meeting of radiologists, urologists, and nephrologists, except for bleeding patients. The criteria for embolization were tumor diameter greater than 4 cm, presence of an aneurysm larger than 5 mm, and/or symptoms such as bleeding due to rAML rupture irrespective of size. All SAE procedures were performed by two experienced interventional radiologists (RL and MM) who were familiar with the procedure.

SAE was performed either on an outpatient basis or during hospitalization, under local anesthesia, through the common femoral artery, using 5-French (Fr) angiographic catheters and coaxial microcatheters. An Allura XD 20 Clarity (Philips, Best, The Netherlands) angiography suite was used for all patients. An aortogram was obtained first through a 5-Fr sheath to locate the renal arteries and to identify any accessory renal arteries or extra-renal feeding arteries. Selective renal angiography was then performed to assess the vascularization of the rAML, extension of tumor vessels outside the normal nephogram, and vessel displacements by the tumor, as well as to identify aneurysms. In patients with severe bleeding, active extravasation and retroperitoneal blood were identified. Next, superselective catheterization of the rAML feeding vessels was achieved using a coaxial 2.0- to 2.7-Fr microcatheter (Progreat^®^; Terumo, Tokyo, Japan) to spare as much renal parenchyma as possible.

The microcatheter was rinsed with 5% dextrose (5%) before the procedure to avoid early polymerization. Glubran^®^2 was combined with ethiodized oil (Lipiodol^®^ Ultra-Fluid; Guerbet, Aulnay-sous-Bois, France). The injection was done under fluoroscopic guidance. A homogeneous NBCA-Lipiodol^®^ mixture was prepared immediately before the injection using two 5 mL luer-lock syringes and a three-way stopcock. A high NBCA dilution of 1:6 was used to increase mixture fluidity, thereby allowing distal embolization (Figure 1). This dilution was arbitrarily chosen based on our experience with tumor devascularization, considering that using a higher dilution does not necessarily allow a more distal embolization. Embolization was performed in free or blocked flow. Effectiveness was assessed visually during SAE, and the injection was stopped when substantial reflux occurred. The microcatheter was then promptly removed.

Technical success was defined as stagnant blood in the feeding arteries and lack of opacification of the rAML on the post-SAE angiogram [18].

A vascular closure device was routinely placed at the puncture site (FemoSeal^®^; Terumo, Tokyo, Japan) for femoral access. Patients were monitored in the interventional unit for the first 2 h and in the ambulatory radiological department for 4 additional hours. They were then discharged home with a prescription for 2 weeks of a prophylactic oral antibiotic and a nonsteroidal anti-inflammatory drug combined with an analgesic.

### 2.4. Follow-Up

All medical records and outpatient charts were revised. Tumor size, percentage of macroscopic fat, and aneurysm size were measured as described above. Follow-up with clinical examinations and laboratory tests was provided by a urologist or nephrologist. CT or MRI was generally performed after 3–6 months then annually. Patients with insufficient follow-up were contacted by telephone to verify the absence of recurrence and to ask for the most recent imaging study and laboratory test results. Recurrence was defined as an increase in tumor size of more than 2 cm on follow-up imaging and/or persistent hypervascular feature and/or recurrent symptoms requiring repeat embolization [19]. Minor and major complications were defined according to Society of Interventional Radiology guidelines [20].

### 2.5. Statistical Analyses

Qualitative variables were described as percentages and quantitative variables as means (SD) or medians (range). As there was no control group, each patient was his or her own control. Estimated tumor volume and creatinine levels before and after SAE were compared by non-parametric univariate analysis using the Wilcoxon rank-sum test. Factors influencing rAML volume reduction were looked for by non-linear univariate regression. The following potential confounding factors (TSC status, age, rAML volume before SAE, % of fat within rAML) have been taken into account in a linear regression model, with a robust estimate of variance. Linearity was checked using fractional polynomial. *p* values < 0.05 were considered significant. The statistical analyses were performed using STATA software (version 14.0, STATA, College Station, TX, USA).

## 3. Results

### 3.1. Patients

We identified 26 patients who underwent SAE with cyanoacrylate glue for rAML at our center during the study period. Among them, two were not included: one patient with multiple large rAMLs at high risk for bleeding underwent embolization of the entire kidney before nephrectomy and another had no available clinical, laboratory, or radiological information obtained before or after SAE. This left 24 patients, with 27 rAMLs, for the study. The diagnosis was made by imaging studies in all but 3 patients, in whom the absence of visible macroscopic fat required a transparietal puncture to obtain a biopsy for histological examination.

We reviewed the medical records using our in-house software to record the demographic and clinical characteristics, TSC status, and number and location of rAMLs. Table 1 reports the main patient characteristics.

### 3.2. Selective Arterial Embolization (SAE)

Table 2 lists the characteristics and outcomes of the endovascular procedure. SAE was technically successful for 25 of the 27 rAMLs. Hemorrhage was stopped in all bleeding rAMLs (Figure 2). Only moderate devascularization was achieved for a hilar rAML with multiple small feeding arteries impossible to catheterize and another tumor received an accessory supply from an adrenal branch that was not completely occluded despite protective placement of a detachable microcoil in the proximal part of the adrenal artery to prevent non-target distal embolization within it with glue. The only rAML that required repeat SAE was partially revascularized by a collateral branch from a lumbar artery arising from the aorta and manifested clinically as flank pain.

Median hospital stay length was 2 days (range, 1–22). Of the 24 patients, seven (29%) returned home on the same day as the SAE, after day-hospital monitoring.

### 3.3. Complications

Minor complications were recorded in 16 patients, of whom 15 had postembolization syndrome (PES) that was managed medically and one had a minor allergic reaction that also resolved with medications.

We recorded major complications in three patients. One patient had a retroperitoneal abscess diagnosed 17 days after PES and successfully treated by drainage. Another experienced a puncture-site hematoma for which emergent CT showed no pseudoaneurysm or active bleeding. The remaining patient was a 77-year-old female transferred to our center’s intensive care unit with massive retroperitoneal bleeding from a large rAML (17.4 cm). She required several blood transfusions for hypovolemic shock. After hemodynamic stabilization, SAE was performed. A pseudoaneurysm at the femoral puncture site was managed percutaneously. Liquefaction and infection of the rAML required surgical drainage. However, she died of hemodynamic failure 20 days after SAE and before the first follow-up imaging; this patient was therefore excluded from the statistical analysis.

### 3.4. Other Outcomes

The patients were followed until January 2021. One patient was lost to follow-up immediately after SAE. The mean clinical follow-up in the study population patients was 19.7 months (range, 1.5–56) and the mean radiological follow-up was 15 months (range, 1–72). No tumor regrowth occurred during this follow-up period. A single rAML (3.7%) was treated surgically during follow-up because of major septic complication.

The reduction in tumor volume was assessed based on the CTs or MRIs obtained before and after SAE. CT was used in 18 and 13 patients before and after SAE, respectively; the corresponding numbers for MRI were 6 and 10, after exclusion of the patient lost to follow-up. The mean tumor volume before and after SAE was 143.3 ± 162.2 mL and 78.8 ± 110 mL, respectively (*p* < 0.0001). The mean volume reduction was 55.1 ± 24.9% (Figure 3).

Creatinine levels were obtained for 18 patients. The mean creatinine levels before and after SAE were 69.3 ± 16.8 µmol/L and 73.4 ± 22.6 µmol/L, respectively, after a mean follow-up of 13.5 months (range, 0–51). The variation was not statistically significant (*p* = 0.27). None of the patients experienced a significant decrease in kidney function following SAE.

### 3.5. Prognostic Factors of Angiomyolipoma Shrinkage

Before SAE, a density histogram was obtained from the CT images (*n* = 18) to determine the percentage of macroscopic fat, defined as attenuation below −20 HU, within the rAMLs. The mean percentage of macroscopic fat was 46.3% (range, 0–91%). This analysis showed that a low-fat content predicted greater volume reduction (*p* < 0.001), in keeping with the greater effect of embolization on the angiomyogenic components of AMLs.

By univariate analysis, greater tumor volume before SAE and longer length of radiological follow-up were significantly associated with a greater decrease in tumor size after SAE (Table 3). In contrast, neither TSC status (patients treated with everolimus) nor age were significantly associated with the reduction in tumor volume.

## 4. Discussion

In our retrospective single-center study of 25 patients with 27 rAMLs, including four treated on an emergency basis for severe bleeding, SAE with cyanoacrylate glue had an excellent technical success rate, with few complications. These good results persisted throughout radiological (15 months) and clinical (19.7 months) follow-up. The reduction in tumor volume was substantial, and no patient experienced a deterioration in kidney function after SAE.

SAE was first performed as a treatment for rAML in 1984 by Adler et al. [21]. SAE is a well-established treatment option for patients with rAMLs that cause bleeding or are at high risk for bleeding. This minimally invasive technique preserves kidney function and has a low complication rate [10,22,23]. However, the optimal embolization material is unclear, as most studies did not provide data on individual agents and/or pooled patients treated with different agents [10]. Liquid agents have the advantage of allowing distal embolization of the entire vascular bed. A study of SAE using ethylene vinyl alcohol (EVOH) to treat 27 rAMLs in 22 patients showed excellent outcomes [12]. However, EVOH causes burning pain during injection into peripheral arteries, requiring either general anesthesia or deep sedation [12]. In addition, EVOH creates specific artifacts that interfere with the evaluation of monitoring CT scans. Cyanoacrylate glue provided encouraging results in two anecdotal case reports [24,25]. We are unaware of cohort studies of cyanoacrylate for rAML treatment. Cyanoacrylate has a number of advantages for lesions supplied by end arteries, such as rAMLs, although there is a learning curve for the operator. The liquid polymerizes rapidly upon contact with blood. However, dilution in Lipiodol decreases the speed of polymerization, thus allowing complete and still rapid devascularization of the vascular bed [16]. In our study, a dilution of 1:6 was used in all cases. The addition of Lipiodol also allows monitoring of the flow of embolic material and detection of possible reflux along the microcatheter. Contrary to microparticles and other liquid agents, cyanoacrylate adheres to the vessel wall, inducing substantial inflammation and remodeling that contribute to lumen obliteration [26]. Another advantage of NBCA/Lipidol mixture is that the fast polymerization as well as the radiopaque feature potentially decreases the risk of nontarget embolization as compared to microparticles. The high technical success rate in our cohort (92.6%) is consistent with the literature [10,27]. We observed no complications specific of NBCA (e.g., non-target embolization, reflux, or microcatheter blockage). However, it is mandatory to check during the angiography the absence of arteriovenous shunts within the rAML although this situation is exceptional. Indeed, in case of shunts, glue can be contraindicated or at least used with caution only after a lipiodol test injection in order to verify the absence of venous passage.

Although rAMLs often are benign and go unnoticed, they tend to grow, eventually becoming symptomatic [28]. Size has long been recognized as a risk factor for rupture. In a 1986 study, among rAMLs larger than 4 cm, 82% were symptomatic and 51% presented with bleeding [29]. This threshold of 4 cm is still used today for indicating prophylactic treatment. Presence of one or more aneurysms also increases the risk of rupture [8]. Before 1976 and the development of cross-sectional imaging, a formal diagnosis was not possible and a malignancy could not be excluded, explaining that total nephrectomy was performed for 90% of sporadic rAMLs [30]. Now, rAMLs are diagnosed simply based on the presence of macroscopic fat by CT or MRI. However, three of our patients required a biopsy due to an atypical fat-free appearance by CT or MRI. A combination of MRI signs has been reported to differentiate fat-poor rAMLs from renal cell carcinoma [31]. The validity of the 4-cm cutoff for deciding to perform prophylactic embolization may be open to criticism. In a review, only slightly more than half the rAMLs responsible for bleeding were larger than 4 cm [29]. In a study of 29 rAMLs, tumor size greater than 4 cm had only 38% specificity for bleeding, compared to 86% for aneurysm size 5 mm or larger [8]. Thus, the 4 cm cutoff for tumor size may have limited relevance in clinical practice. However, in a study of 34 rAMLs, complete disappearance of the mass was achieved for fat-poor tumors smaller than 4 cm [32].

The volume reduction in our cohort was significant and within the reported range of 43% to 75% [27,33,34]. Greater volume reduction was associated with a smaller tumor fat component. A retrospective study of 19 patients with 39 rAMLs demonstrated significantly greater volume reduction in tumors containing less than 50% of fat (84% vs. 50%; *p* < 0.00001) [27]. In the multivariate analysis, the model that best predicted volume reduction included only the percentage of fat content (R^2^ = 0.61; *p* < 0.0001). A density histogram study of 34 rAMLs also found that the volume reduction was much smaller when the fat component was large [35]. In a study of 34 rAMLs, complete disappearance of the mass was achieved for fat-poor tumors smaller than 4 cm. Thus, embolization chiefly decreases the angiomyogenic component, which is strongly linked to the risk of hemorrhagic rupture [8,32,36,37]. Consequently, the global decrease in tumor volume may not be the best measure of embolization efficacy, and the decrease in the non-fat component may deserve evaluation as a possibly more reliable efficacy criterion. Longer follow-up was associated with greater tumor volume reduction in our patients. In another study, most of the reduction occurred during the first year, with further smaller reductions to a plateau at 3 years, and the authors suggested that small volume decreases within the first year might predict regrowth in the long term [38]. In our study, the only patient who required repeat embolization due to a relapse had the smallest volume reduction (1.6%). Volume reduction did not correlate significantly with TSC status or age.

The most frequent complication in our study was PES, an inflammatory reaction causing pain and fever within 48 h after SAE, but unrelated to the type of embolic agent used. Nonsteroidal anti-inflammatory agents or corticosteroids are effective against these symptoms. In a meta-analysis of 30 studies, PES occurred in 54% of cases, in keeping with our data [23]. Lower frequencies of 6% to 12.5% were observed in studies that involved routine nonsteroidal anti-inflammatory drug therapy [7,39]. Only three of our patients experienced severe complications. A patient with a puncture-site hematoma but no pseudoaneurysm was successfully treated by compression and monitoring. Another patient developed a retroperitoneal collection, which was successfully drained. The third patient, who presented with hypovolemic shock due to massive retroperitoneal rupture of the tumor, developed a false aneurysm of the common femoral artery, which was successfully treated using a percutaneous approach, followed by liquefaction and infection of the embolized rAML, which was treated surgically. She died of decompensated heart failure 20 days after SAE. Liquefaction and abscess formation after SAE are secondary to ischemic necrosis of the tumor and can occur regardless of the embolization agent used [10,11,23,32,33,40]. A systematic review of rAML outcomes showed a 6.9% morbidity rate after SAE, which was lower than the 12% rate after partial nephrectomy [10,41]. A single (4.2%) patient required nephrectomy in the SAE suite, in keeping with the literature [7]. The low complication rate was accompanied with a short hospital stay length (median, 2 days vs. 6 days for nephron-sparing surgery) [41]. Nearly a third of our patients had SAE on a day-hospital basis, giving this treatment technique a clear cost advantage over surgery. Kidney function was preserved following SAE, as described previously [7,23,27,32,36,42,43]. The preservation of the parenchyma and therefore of organ function is particularly important in TSC patients in whom involvement is most often bilateral. None of our patients experienced bleeding after the first embolization.

The single patient who required repeat SAE, after 14 months, had flank pain that led to the diagnosis of partial tumor revascularization by a collateral artery arising from the aorta. This second procedure improved the pain and increased the tumor devascularization seen on follow-up imaging studies. However, other studies with longer follow-up times found higher rates of repeat SAE [7,10]. After 57.6 months, the proportions of patients without repeat SAE were 71% (95% CI, 55–87%) after 5 years and only 37% (95% CI, 17–58%) after 10 years [7]. Factors significantly associated with repeat SAE were failure of initial SAE (*p* = 0.02), larger rAML size (*p* = 0.07), and bilateral rAMLs (*p* = 0.07).

All 5 patients with TSC in our study were on everolimus therapy at the time of SAE. This targeted therapeutic selectively stops tumor progression and promotes tumor regression by selectively inhibiting the mTORC-1 pathway. In the EXIST-2 randomized controlled double-blind trial in patients with TSC and rAML, everolimus therapy decreased non-embolized rAML volume by at least 50% in 42% of patients [44,45].

The main limitations of this study are the retrospective single-center design, small number of patients, and heterogeneous presentations. Thus, about a fifth of our patients had TSC and about four-fifths underwent prophylactic SAE as opposed to emergent SAE. Second, the use of everolimus in all five patients with TSC may introduce bias, as this drug can decrease rAML volume over time [44,45]. However, our adjustment in regression model analysis avoided confounding bias in TSC patients treated with everolimus. It means that TSC status had not impact on embolized rAML volume independently. Third, the mean follow-up was relatively short (15 months radiologically) and varied widely (1–72 months). However, as pointed out above, most of the volume reduction after SAE is achieved during the first year [38]. Finally, the frequency of clinical follow-up and laboratory tests varied across patients, and 6 of the 24 patients had incomplete creatinine data. Further studies with longer follow-ups and comparisons of embolic agents are needed.

## 5. Conclusions

SAE of rAMLs using a mixture of NBCA and Lipiodol is feasible, safe, and effective for the emergency treatment of bleeding and for bleeding prophylaxis. In our patients, SAE had an excellent technical success rate with low complication and relapse rates and no rebleeding. Major complications were rare and the only patient who died had a very severe presentation and preexisting risk factors. Greater volume reduction was associated with a smaller fat component, a larger initial tumor volume, and longer follow-up.

## Figures and Tables

**Figure 1 jcm-10-04062-f001:**

Prophylactic selective arterial embolization of a large right renal angiomyolipoma (rAML). (**a**) Angiogragram demonstrates two feeding arteries to the hypervascular angiomyolipoma. (**b**) Final angiographic control after superselective embolization of the two arterial branches with a Glubran^®^2/Lipiodol^®^ mixture in a 1:6 ratio. (**c**) Computed tomography scan at the portal phase 2 weeks later showing lipiodol distribution within the rAML and confirming targeted embolization with no more enhancement of the rAML.

**Figure 2 jcm-10-04062-f002:**
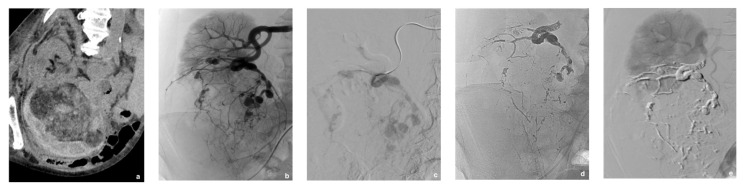
A 77-year-old patient hospitalized for right acute flank pain, hematuria and deglobulisation. (**a**) Unenhanced computed tomography (CT) scan revealed right perirenal hematoma caused by a large right lower pole renal angiomyolipoma (rAML). (**b**) Embolization to stop bleeding was performed urgently. Selective injection of a lower unique branch of the right renal artery showed hypervascularization of rAML with neovascularization and aneurysms. (**c**,**d**) Selective embolization of rAML with a mixture of 1:6 cyanoacrylate glue and lipiodol was performed allowing both distal and proximal embolization of the feeding artery and aneurysms with the same embolic agent. (**e**) Post-procedure angiogram showed total exclusion of rAML with good preservation of viable renal parenchyma.

**Figure 3 jcm-10-04062-f003:**
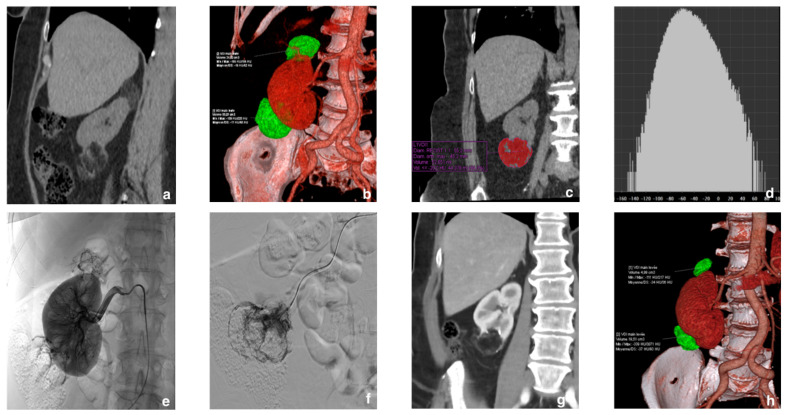
Example of a 58-year-old woman with no history of tuberous sclerosis. (**a**) Two fat-rich renal angiomyolipomas (rAMLs) were discovered in the right kidney during computed tomography (CT) done for other reasons. (**b**) The volume-rendering reconstruction shows the two right polar rAMLs. (**c**,**d**) Coronal CT image with tumor segmentation and a density histogram: note the large fat component of the lower polar lesion, estimated at 84.3%. (**e**) Selective prophylactic arterial embolization was decided. The right renal angiography shows the rAMLs at the upper and lower poles. (**f**) After superselective catheterization and angiography of both rAMLs, the injection of a mixture of Glubran^®^2 and Lipiodol in a 1:6 ratio achieved complete devascularization of both tumors. (**g**,**h**) Contrast-enhanced control CT at 17 months with volume rendering showed an excellent result with 79.8% and 64% volume reductions of the upper and lower rAMLs, respectively.

**Table 1 jcm-10-04062-t001:** Mean features of the 24 study patients (27 renal angiomyolipomas (rAMLs)).

Variable	Data
No. of patients	24
No. of embolized rAMLs	27
Females/Males, *n* (%)	22 (91.6)/2 (8.3)
Age, y, mean ± SD	51 ± 20.8
Tuberous sclerosis, *n* (%)	5 (20.8)
Everolimus treatment, *n* (%)	5 (20.8)
Indication of SAE, *n* (%)	
Prophylactic SAE: prevention of bleeding ^a^	20 (83.3)
Bleeding requiring emergency treatment	4 (16.7)
Clinical presentation of rAML, *n* (%)	
Shock	3 (12.5)
Retroperitoneal bleeding	4 (16.7)
Hematuria	2 (8.3)
Pain	13 (54.2)
No symptoms	11 (45.8)
Multiple rAMLs, *n* (%)	7 (29.2)
Bilateral rAMLs, *n* (%)	7 (29.2)
Presence of aneurysms, *n* (%)	9 (33.3)
Side of embolized rAMLs, *n* (%)	
Right	14 (51.9)
Left	13 (48.1)
Location of embolized rAMLs	
Central	2 (7.4)
Exophytic	25 (92.6)

SAE, selective arterial embolization; rAML, renal angiomyolipoma; y, year; *n*, number; %, percentage; SD, standard deviation; ^a^ tumors more than 4 cm in size, aneurysm more than 5 mm in size, and/or non-emergent symptoms.

**Table 2 jcm-10-04062-t002:** Outcomes, follow-up, kidney function, and tumor size after selective arterial embolization (SAE) of one or more renal angiomyolipomas (rAMLs).

Variables	Data
rAMLs before SAE	
Size, cm, mean ± SD	8.0 ± 2.9
Size, cm, median (range)	7.6 (4.2–15.4)
Volume, cm^3^, mean ± SD	143.3 ± 162.2
Volume, cm^3^, median (range)	82.3 (21.0–646.0)
Radiologic follow-up, months, mean (range)	15 (1–72)
rAMLs after SAE (at mean follow-up)	
Size, cm, mean ± SD	6.1 ± 3.0
Size, cm, median (range)	5.0 (2.5–13.1)
Volume, cm^3^, mean ± SD	78.8 ± 110.0
Volume, cm^3^, median (range)	28.1 (2.7–348.0)
Tumor size reduction at mean follow-up	
cm, mean ± SD	1.9 ± 1.4
%, mean ± SD	25.2 ± 16.0
Tumor volume reduction at mean follow-up	
cm^3^, mean ± SD	64.5 ± 69.0
%, mean ± SD	55.1 ± 24.9
Serum creatinine level	
Before SAE, µmol/L, mean ± SD	69.3 ± 16.8
After SAE, µmol/L, mean ± SD	73.4 ± 22.6
Technical success ^a^ rate, *n* (%)	25 (92.6)
Need for re-embolization, *n* (%)	1 (3.7)
Renal surgery after SAE, *n* (%)	1 (3.7)
Minor complications within 1 month, *n* (%)	16 (61.5)
PES	15 (57.7)
Allergic reaction	1 (3.7)
Major complications within 1 month, *n* (%)	3 (11.1)
Pseudoaneurysm	1 (3.7)
Abscess	2 (7.4)
Death	1 (3.7)

SAE, selective arterial embolization; rAML, renal angiomyolipoma; PES, postembolization syndrome; SD, standard deviation; cm, centimeter; %, percentage; µmol/L, micromoles per liter; ^a^ immediate and complete devascularization on follow-up arteriography.

**Table 3 jcm-10-04062-t003:** Factors significantly associated with the decrease in tumor volume after selective arterial embolization.

Variables	*p* Value
Aging	0.054
Lower % of fat	0.001
Greater initial volume	0.014
Longer of follow-up time	0.0001
Tuberous sclerosis complex	0.059

%, percentage; *p* < 0.05 was considered as statistically significant.

## Data Availability

All the study data are reported in this article.

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
