# Peer review of "Selective Arterial Embolization of Renal Angiomyolipomas with a N-Butyl Cyanoacrylate-Lipiodol Mixture: Efficacy, Safety, Short- and Mid-Term Outcomes"

_jcm, 2021, doi:10.3390/jcm10184062_

Round 1

Reviewer 1 Report

The authors describe efficacy, safety, and outcomes of selective arterial embolization for rAML with NBCA-lipiodol.
The manuscript is well written and easily readable, contained something new. I feel this manuscript has a potential value for readers involved in this field. However, I think it’s better to improve two points.

First, table 2 may be made a little easier to read. So, sub-items like “rAMLs before SAE”, “rAMLs after SAE” or the like can be underlined.

Second, rAML can be rarely accompanied by arterio-venous shunts, in which case NBCA cannot be used. However, it is common for interventional radiologists that NBCA cannot be used in cases with shunts. So, it might be not necessary to describe it.

Author Response

Responses to Reviewer 1 Comments

The authors describe efficacy, safety, and outcomes of selective arterial embolization for rAML with NBCA-lipiodol.
The manuscript is well written and easily readable, contained something new. I feel this manuscript has a potential value for readers involved in this field. However, I think it’s better to improve two points.

Reply: Thank you very much for your comments. The manuscript has been improved as suggested.

First, table 2 may be made a little easier to read. So, sub-items like “rAMLs before SAE”, “rAMLs after SAE” or the like can be underlined.

Thank you very much for your comments. We fully agree. The table 2 has been made easier to read and understand as suggested. Some lines have been emphasized.

Second, rAML can be rarely accompanied by arterio-venous shunts, in which case NBCA cannot be used. However, it is common for interventional radiologists that NBCA cannot be used in cases with shunts. So, it might be not necessary to describe it.

Reply: Thank you very much for your comments. We fully agree. It is mandatory to check during the angiography the absence of arteriovenous shunts within the rAML although this situation is exceptional. Indeed, in case of shunts, glue can be contraindicated or at least used with caution only after a lipiodol test injection in order to verify the absence of venous passage. It has been described in the discussion section as a new paragraph as suggested.

Reviewer 2 Report

The authors showed effectiveness of SAE for AMLs by NBCA/lipiodol. It is interesting study. However, manuscript design should be reconsider on the issues of prophylactic/hemostasis, short term result/mid-term result, and patient inclusion criteria (remove the patients who receive Everolimus).

Introduction: OK

Materials and Methods:

L130: “A high NBCA dilution of 1:6 was used to increase mixture fluidity”

How did the authors decide the NBCA/lipiodol concentration? Why 1:6, not 1:7 or 1:8?

Results:

L183: Table 1 is hard to understand quickly.

There are 5 patients of TSC patients with Everolimus treatment in the Table 1. The patients who receive Everolimus treatment should be excluded from this study. Because AML will reduce in size by Everolimus only. Then true reduction rate of SAE is hard to evaluate.

In the table 1, 20 prophylactic SAE and 4 SAE for bleeding AML are included. In this study authors describe only the effect of prophylactic SAE is better to show the effect of author’s technique. Then, 4 SAE for bleeding AML should be excluded or described separately.

L191: “another tumor received an accessory supply from an adrenal branch that was not completely occluded despite placement of a microcoil”

Did authors use microcoil in addition to NBCA/lipiodol ? Please mention about the all embolic agent used for SAE.

L193: “partially revascularized by a collateral branch arising from the aorta”

Please describe the name of collateral branch. Usually it should be lumber artery, inferior phrenic artery or middle adrenal artery.

Line 197-8: Table 2

In the table, the authors show “Radiologic follow-up, months, mean (range) 15 (1-72)”. However, as authors described in the discussion session (L330), reduction rate is different in each time course, range of 1-72 months is too wide to see the effect of SAE. Then, the authors should be focus on the short term result (about 1 year after SAE) or mid-term result (about 3 year after SAE) or both, separately.

“False aneurysm” may be “pseudoaneurysm”

Discussion:

L289: In materials and methods, authors described they used 1:6 NBCA/lipiodol. However, in discussion, the authors described “a dilution of 1:6 was used in 92% of cases”. What was used in 8% cases? Please clarify in the materials and methods section.

Author Response

Responses to Reviewer 2 Comments

The authors showed effectiveness of SAE for AMLs by NBCA/lipiodol. It is interesting study. However, manuscript design should be reconsider on the issues of prophylactic/hemostasis, short term result/mid-term result, and patient inclusion criteria (remove the patients who receive Everolimus).

Reply: Thank you very much for your comments. We fully explain our point of view about these interesting comments below in each reply. The manuscript has been improved as much as possible as suggested but unfortunately some suggestions could not be satisfied as expected for study design reasons.

Introduction: OK

Reply: Thank you very much for your comments.

Materials and Methods:

L130: “A high NBCA dilution of 1:6 was used to increase mixture fluidity”

How did the authors decide the NBCA/lipiodol concentration? Why 1:6, not 1:7 or 1:8?

Reply: Thank you very much for your comments. We fully agree that there is no strong recommendation regarding the NBCA/Lipiodol dilution to be used in such a setting. This 1:6 ratio is based on our own but extensive experience with the use of glue for tumor devascularization. The most common and normal ratio is considered around 1:3 for most indications. When a distal embolization for tumor bed devascularization is required, the use of a higher ratio is needed but it is admitted that using more than 1:6 or 1:7 ratio does not add anything in terms of dilution. It has been clarified in the line 130 as suggested.

Results:

L183: Table 1 is hard to understand quickly.

Reply: Thank you very much for your comments. The table 1 has been made easier to understand as suggested.

There are 5 patients of TSC patients with Everolimus treatment in the Table 1. The patients who receive Everolimus treatment should be excluded from this study. Because AML will reduce in size by Everolimus only. Then true reduction rate of SAE is hard to evaluate.

Reply: Thank you very much for your comments. First, we don’t fully agree on this point. Indeed, the use of Everolimus can have impact on the size reduction of rAMLs but this impact only occurs for native rAMLs which have not been yet embolized, due to the vascular mechanism of action of Everolimus. When rAMLs are treated with embolization and so devascularized, the impact of Everolimus is almost nil on these rAMLs on the long-term and Everolimus does not interfere in the size reduction of these embolized rAMLs which is then only due to the effect of embolization. We then consider that excluding these TSC patients form the study would not be relevant and would probably not change anything regarding the volume of rAMLs after embolization despite Everolimus treatment which mainly acts on non-embolized rAMLs. No trial exists to show any additional action of Everolimus on embolized rAMLs. Second, our study, although being the largest one to date with the use of glue in such a setting, contains overall only 24 patients which represents a small sample. Excluding the 5 TSC patients would even reduce this sample size, leading to lack of power, which would be a shame. For these reasons, we feel preferable to keep the number of patients as it is. However, we remain at your disposal to make required changes if you think them relevant. Some additional sentences have been added in the limitation section of the discussion to explain more in details the role of Everolimus in such a setting.

In the table 1, 20 prophylactic SAE and 4 SAE for bleeding AML are included. In this study authors describe only the effect of prophylactic SAE is better to show the effect of author’s technique. Then, 4 SAE for bleeding AML should be excluded or described separately.

Reply: Thank you very much for your comments. We are not sure to understand this comment and the interest to exclude the 4 SAE for bleeding rAMLs. Indeed, we describe in our study the effect of both prophylactic and emergent embolization of rAMLs, not only the effect of prophylactic. Overall, 24 patients were included, 20 asymptomatic patients for prophylactic SAE and 4 bleeding patients for emergent SAE. The effect of SAE on volume size reduction as well as the technical/clinical success were reported for all patients, even for bleeding patients. It is well described in the materials and methods section. Furthermore, excluding 4 more patients (the bleeding ones), in addition to the potential 5 TSC patients, would lead to exclude 9 of the 24 patients at the end, which appears to us not reasonable for the flow of the paper, in terms of sample size, and for readers. Again, we remain at your disposal to discuss these points deeper if you think them relevant.

L191: “another tumor received an accessory supply from an adrenal branch that was not completely occluded despite placement of a microcoil”

Did authors use microcoil in addition to NBCA/lipiodol ? Please mention about the all embolic agent used for SAE.

Reply: Thank you very much for your comments. This is a transcription error. In fact, we used coils in one case for protecting proximally the adrenal artery before injection of the NBCA/Lipiodol mixture more distally in a branch of this adrenal artery which was impossible to catheterize. The goal was to prevent distal embolization of the entire adrenal artery. So, coils here were not used for embolization but for protection. It was misreported. The sentence has been changed in the materials and methods section as suggested.

L193: “partially revascularized by a collateral branch arising from the aorta”

Please describe the name of collateral branch. Usually it should be lumber artery, inferior phrenic artery or middle adrenal artery.

Reply: Thank you very much for your comments. The name of the collateral branch has been described in the materials and methods section as suggested. It was indeed a branch of a lumbar artery.

Line 197-8: Table 2

In the table, the authors show “Radiologic follow-up, months, mean (range) 15 (1-72)”. However, as authors described in the discussion session (L330), reduction rate is different in each time course, range of 1-72 months is too wide to see the effect of SAE. Then, the authors should be focus on the short-term result (about 1 year after SAE) or mid-term result (about 3 year after SAE) or both, separately.

“False aneurysm” may be “pseudoaneurysm”

Reply: Thank you very much for your comments. Indeed, this is a limitation of the study due to its retrospective design, as already mentioned in the limitations section of the discussion. The mean radiological follow-up was relatively short (15 months) but not that much, and varied widely (1-72 months). In fact, this mean follow-up can be considered not as “short” but as “intermediate” (mid), short follow-up being usually reported within 1 month. In addition, as pointed out in the discussion, most of the volume reduction after SAE is achieved during the first year. Having a 3-year follow-up would probably not allow any additional information regarding the size reduction. Last, the frequency of radiological follow-up varied across patients and the last radiological follow-up was the best assessable endpoint in the present retrospective study despite limitations. That’s why we reported short-term outcomes (within 1 month) concerning complications as usually reported in the literature, and mid-term outcomes (mean follow-up of 15 months) mainly concerning tumor size reduction. It has been clarified in table 2. 

Furthermore, “false aneurysm” has been changed for “pseudoaneurysm” in the table 2 and through all manuscript, as suggested.

Discussion:

L289: In materials and methods, authors described they used 1:6 NBCA/lipiodol. However, in discussion, the authors described “a dilution of 1:6 was used in 92% of cases”. What was used in 8% cases? Please clarify in the materials and methods section.

Reply: Thank you very much for your comments. Indeed, this is a mistake. I confirm that a dilution of 1:6 (NBCA/Lipiodol) was used in all cases (100%) as extensively described in the materials and methods section. It has been corrected in the discussion section as suggested.

Round 2

Reviewer 2 Report

Thank you very much for your quick response and modification of your manuscript.

One thing I would like to discuss with the authors is how to treat the patients with Everolimus therapy.

The authors think tumor reduction effect of Everolimus therapy occurs only for native renal AMLs (AMLs without SAE) and does not occur renal AMLs after SAE. However, there are the articles that renal AMLs regrow after SAE in TSC patientsa,b and Everolimus reduces the size of regrown AMLs (mean reduction rate is 53%)c.

Then, 5 TSC patients with Everolimus therapy should be excluded from this study to see only the effect of SAE.

references

a. Ewalt DH, Diamond N, Rees C, et al (2005) Long term outcome of transcatheter embolization of renal angiomyolipomas due to tuberous sclerosis complex. J Urol 174:1764–1766

b. Sheth RA, Feldman AS, Paul E, et al (2016) Sporadic versus tuberous sclerosis complex-associated angiomyolipomas: predictors for long-term outcomes following transcatheter embolization. JVIR 27:1542-1549

c. Hatano T, Matsu-ura T, Mori K, et al (2018) Effect of everolimus treatment for regrown renal angiomyolipoma associated with tuberous sclerosis complex after transcatheter arterial embolization. Int J Clin Oncol 23:1134-1139

Author Response

Responses to Reviewer 2 Comments

Thank you very much for your quick response and modification of your manuscript.

One thing I would like to discuss with the authors is how to treat the patients with Everolimus therapy.

The authors think tumor reduction effect of Everolimus therapy occurs only for native renal AMLs (AMLs without SAE) and does not occur renal AMLs after SAE. However, there are the articles that renal AMLs regrow after SAE in TSC patientsa,b and Everolimus reduces the size of regrown AMLs (mean reduction rate is 53%)c.

Then, 5 TSC patients with Everolimus therapy should be excluded from this study to see only the effect of SAE.

References

  1. Ewalt DH, Diamond N, Rees C, et al (2005) Long term outcome of transcatheter embolization of renal angiomyolipomas due to tuberous sclerosis complex. J Urol 174:1764–1766
  2. Sheth RA, Feldman AS, Paul E, et al (2016) Sporadic versus tuberous sclerosis complex-associated angiomyolipomas: predictors for long-term outcomes following transcatheter embolization. JVIR 27:1542-1549
  3. Hatano T, Matsu-ura T, Mori K, et al (2018) Effect of everolimus treatment for regrown renal angiomyolipoma associated with tuberous sclerosis complex after transcatheter arterial embolization. Int J Clin Oncol 23:1134-1139

Reply:  Thank you very much for your comments. We fully agree after reviewing of the literature. We also agree that TSC status (meaning patients with everolimus treatment) could be a confounding factor, having concomitant impact on the embolized rAML volume. That’s exactly why the potential effect of TSC status/everolimus treatment was taken into account in the initial analysis by adjustment in the statistical regression model. The results showed that it had no independent effect on rAML volume reduction after SAE (Table 3). Maybe it was not clearly expressed in the statistical analysis paragraph. We clarified it for more understanding. This statistical approach was specifically used by our statistician for emphasizing these results and avoiding to exclude TSC patients from the study that could have led to a loss of statistical power for other variables (Table 3). Adjustment is exactly and usually made for this purpose from a statistical point of view (see references: Werner Vach et al. “Regression models as a tool in medical research” and Altman et al. “Practical statistics for medical research”). It has been clarified in the materials and methods section and the limitations section of the manuscript.